# Unique reactivity of nanoporous cellulosic materials mediated by surface-confined water

Marco Beaumont [1,2✉], Paul Jusner [1], Notburga Gierlinger [3], Alistair W. T. King [4], Antje Potthast [1], Orlando J. Rojas[2,5] & Thomas Rosenau [1,6✉]

The remarkable efficiency of chemical reactions is the result of biological evolution, often involving confined water. Meanwhile, developments of bio-inspired systems, which exploit the potential of such water, have been so far rather complex and cumbersome. Here we show that surface-confined water, inherently present in widely abundant and renewable cellulosic fibres can be utilised as nanomedium to endow a singular chemical reactivity. Compared to surface acetylation in the dry state, confined water increases the reaction rate and efficiency by 8 times and 30%, respectively. Moreover, confined water enables control over chemical accessibility of selected hydroxyl groups through the extent of hydration, allowing regioselective reactions, a major challenge in cellulose modification. The reactions mediated by surface-confined water are sustainable and largely outperform those occurring in organic solvents in terms of efficiency and environmental compatibility. Our results demonstrate the unexploited potential of water bound to cellulosic nanostructures in surface esterifications, which can be extended to a wide range of other nanoporous polymeric structures and reactions.

[1] Department of Chemistry, Institute of Chemistry for Renewable Resources, University of Natural Resources and Life Sciences Vienna (BOKU), Tulln, Austria. [2] Department of Bioproducts and Biosystems, School of Chemical Engineering, Aalto University, Aalto, Finland. [3] Institute for Biophysics, Department of Nanobiotechnology, University of Natural Resources and Life Sciences, Vienna, Austria. [4] Materials Chemistry Division, Department of Chemistry, University of Helsinki, Helsinki, Finland. [5] Bioproducts Institute, Departments of Chemical and Biological Engineering, Chemistry and Wood Science, University of British Columbia, Vancouver, BC, Canada. [6] Johan Gadolin Process Chemistry Centre, Åbo Akademi University, Turku, Finland. ✉email: marcobeaumont1@gmail.com; thomas.rosenau@boku.ac.at

Water, as the solvent of life, is ubiquitous in the chemical processes of living organisms[1], which thrive only in aqueous systems. Water controls the activity of enzymes, acts as a nucleophile and mediates charge and proton transfer in biological reactions. In the presence of water, nature enables the miraculous array of complex biochemical reactions at high efficiency and under mild conditions. These also include processes which are less favoured in the presence of water[2], such as dehydration and esterification. They do nevertheless occur due to presence of optimised enzymes[3]; and the fact that water is spatially confined in cells and organelles[1,4]. Confined water is restricted in mobility and features an anomalous behaviour, if compared to the fluid in bulk, this directly affects its chemical and physical properties[5,6]. Exploiting the potential of this singular water state for chemical reactions has led to very important scientific contributions[7], which take advantage of confined water, for example, in artificial porous solids[8–12] or biphasic fluid systems[13–15]. Current limitations are the restrictions to small molecules as reactants, the use of organic solvents and the rather complex design of these reaction systems. Expanding this concept to naturally occurring and sustainable materials, which confine water due to their intrinsic native structure, would be a major contribution to the field. This is especially true in times of climate[16] and plastic crises[17]. This renders the development of sustainable reactions, and materials, most timely and necessary.

Nature's best polymeric fibres are essential structural components in biological tissues: keratin[18], collagen[19] and fibroin[20,21] as examples of protein fibres, along with cellulose[22] and chitin[23], as representatives of structured polysaccharide. All these materials are based on nano-scaled assemblies, covered by surface-confined water under ambient conditions. Cellulose fibres are used herein as representative example for this class of hierarchically structured, nanoporous biopolymers. Cellulose[24], from the molecular viewpoint, has a rather simple chemical structure based on β-O-1,4-linked glucopyranose repeating units that are assembled into rigid nanofibres, i.e. elementary fibrils, as the smallest subunits (Fig. 1).

The cellulose nanofibres are composed of inter-connected, axially twisted crystallites[25] with a chemically accessible and hydrophilic surface[26]. The nanoporosity of cellulose originates from this interfibrillar assembly and structural defects that arise from the spatial mismatch and dislocations of individual nanofibers and assembled bundles, enabling solvent interpenetration and fibre swelling[27]. The abundance of surface hydroxyl groups in this hierarchical structure endows hygroscopicity and sorption-induced swelling of cellulose fibres[28,29]. Main contributors in the swelling are the surfaces of the individual nanofibers[30,31], that are covered with hydration layers, under ambient conditions[26,28]. Due to the strong cellulose–water interactions, this surface-bound water can be distinguised from bulk water by its properties and it is referred to as freezing or non-freezing (bound) water[32–35]. Non-freezing water is directly situated at the nanofibre surface, whereas freezing water accumulates in interfibrillar pores[32]. Interestingly, the omnipresent water has mostly been considered as a major hindrance in cellulose chemistry, when it comes to typical organic chemistry reactions. The presence of water makes reactions either unfavourable or would act as competitor to surface hydroxyl groups. As a consequence, typical reactions have been mainly performed under anhydrous conditions[36–38]. This is in stark contrast to recent efforts to use water as reaction promoter[39,40].

With this work, we hope to induce a paradigm shift in materials chemistry by exposing the potential of confined water to promote surface reactions of nanoporous polymeric structures. This principle is demonstrated here using as example the acetylation of cellulose fibres, as representative as nanoporous systems[28].

## Results

**Surface-confined water as reaction medium.** Cellulosic fibres of high purity (92% cellulose content) were used as starting material. Due to their hierarchical and nanoporous structure, they can confine up to 0.5 mL of water per gram of fibre (fibre saturation point, Supplementary Table 1). In the case of the cellulosic fibre, cf. Figure 1, only the surface hydroxyl groups of the elementary fibrils are accessible for chemical reactions and, unless noted otherwise, all experiments were conducted using fibres with equilibrated moisture content (EMC) of 7 wt% (at 50% relative humidity and room temperature, Supplementary Table 1). The number of water molecules is in the same stoichiometric range as that of the cellulose´s surface hydroxyl groups and hence water is strongly bound to the surface. In this work, we used solely the solid reactant, N-acetylimidazole[41–44] and a catalyst, imidazole, to acetylate the cellulose surface. Imidazole was chosen because of its structural analogy to histidine, its associated enzyme-like catalytic behaviour[45] and affinity for cellulose surfaces[46]. The heterogeneous reaction of cellulose and N-acetylimidazole commenced instantly upon solid-state mixing (Fig. 2A). Efficient mixing was realised by using short vibratory ball milling. In contrast to conventional (mechano)chemical esterifications[47–49], acetylation occurred promptly, with no need for continuous energy supply in the form of milling (or heating). Earlier reports on (mechano)chemical esterifications of cellulose and other polymers have relied on the presence of liquid/solution media (chemicals), such as acetic anhydride. To the best of our knowledge, our presented approach is the only true solid-state polymer acetylation method available thus far.

Ball milling was chosen as it allows under the selected conditions mild mixing of cellulose with the solid reactants. In contrast to previously published results[50], neither depolymerisation nor decrystallization occurred (Supplementary Fig. 1). We studied the solid-state acetylation in the presence of different water amounts—dry, 7, 20 and 30 wt%—with respect to kinetics, reaction efficiency and regioselectivity. The calculated reaction efficiency was expressed as the ratio of the degree of substitution (DS) and the molar amount of consumed N-acetylimidazole. Since water is conventionally regarded as a nuisance factor in

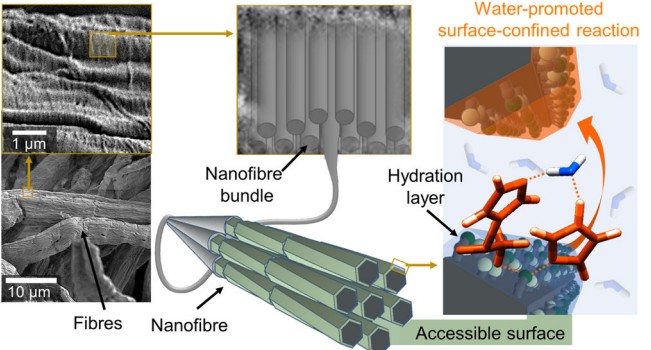

**Fig. 1 Hydration layers in nanoporous polymers as reaction media.** Cellulose, as representative example of a hierarchically structured biopolymer, is composed of individual nanofibres (i.e. elementary fibrils), organised into micro-scaled fibres with nanopores and confined spaces that originate from their interfibrillar assembly. Only the surface of these fibers is chemically accessible (accessible OH-groups are shown as light and dark green beads) and, under ambient conditions, is covered with a hydration layer comprising surface-confined water. This water is explored as nanomedium to promote chemical reactions.

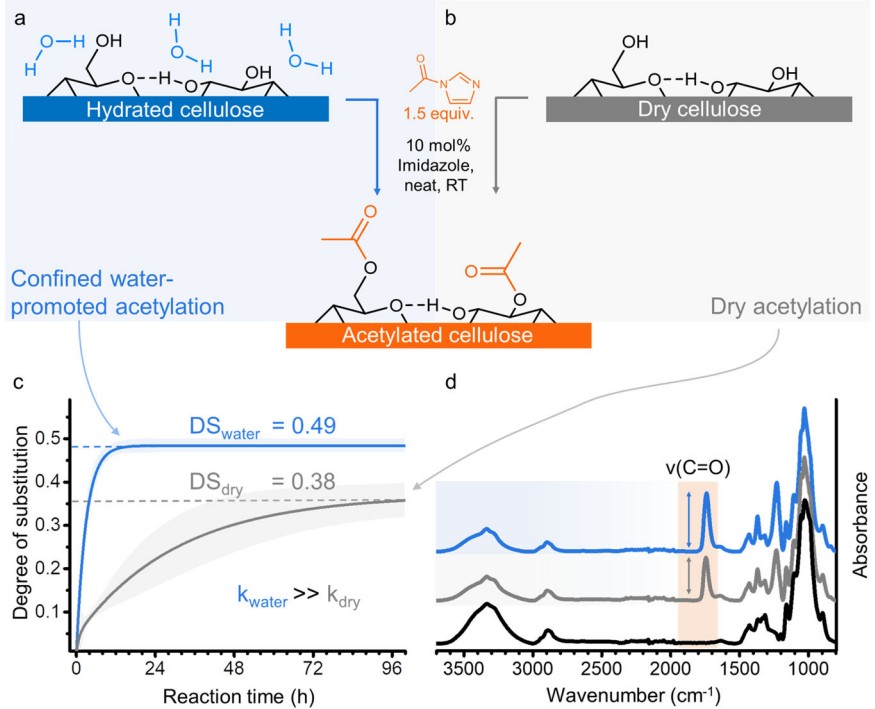

**Fig. 2 Acetylation under the confinement of hydration layers.** Cellulose fibres reacted under solvent-free and heterogeneous (solid-phase) conditions with *N*-acetylimidazole in presence of 7 wt% confined water (**a**, moist cellulose) and dry condition (**b**). The kinetics (**c**) and the infra-red spectra (**d**) show that water, paradoxically, is a key component for the reaction to occur and increases reaction rate and efficiency (**c**, 95% confidence intervals are shown in shaded colour).

esterifications, decreasing the efficiency of this type of reaction, we anticipated the highest reaction efficiency for the dry sample. Unexpectedly, we observed the contrary: the presence of 7 wt% water in the fibre increased not only the reaction rate but also its efficiency (Fig. 2c). The kinetic scatter plots were exponentially fitted and resembled the shape of typical diffusion-controlled reactions[51]. Considering our results, the solid reactants, and the solvent-free reaction conditions, we assume a reaction mechanism based on diffusion through the fibre hydration layers. The reaction is also auto-catalytic as additional formed imidazole (formed after hydrolysis or esterification of *N*-acetylimidazole), can act as a base catalyst[45]. This has the effect of accelerating reaction rates inside the fibre, as the reaction proceeds (Supplementary Fig. 2). Increasing the water content in the cellulose sample further, to 20 wt% and 30 wt%, compared to the equilibrated moisture of 7 wt%, did not significantly influence the kinetics of the reactions (Supplementary Fig. 3) but reduced the DS of the acetylated fibres and hence the overall reaction efficiency (Supplementary Table 2). It is important to point out that in all samples the water was structurally confined to the fibres, since even a 30 wt% water content was below the fibre saturation point (Supplementary Table 1).

**Control of chemical accessibility and effect on bulk properties.** Our results, based on diffusion-edited ¹H-NMR spectra of the acetylated fibres (Fig. 3a) in the ionic liquid-electrolyte [P4444][OAc]:DMSO-d6[52,53] (details can be found in Supplementary Methods and Supplementary Fig. 17–20), showed that smaller amounts of secondary hydroxyl groups were modified at a water content above EMC, i.e. increasing selectivity for the primary 6-OH.

The regioselectivity of the reaction can thus essentially be controlled through the water content. Increasing it to 30 wt% elevated the selectivity towards the primary C6-hydroxyl group to

89%, yielding a fibre with a DS of 0.20 ± 0.04. The number of modified C6-groups is in the range of the value (DS = 0.25) estimated for full surface coverage, assuming a 24-chain elementary fibril model and the crystallite size of the fibre (Supplementary Table 1). This low DS can be explained by aligning our reasoning with previous studies of the acetylation of tyrosine in enzymes, stating that certain 'buried' hydroxyls are not chemically accessible due to hydrogen bonding[54]. We conclude that increasing the number of water molecules at the cellulose surface decreases the chemical accessibility of all hydroxyls but the secondary hydroxyls in particular, due to water–hydroxyl interactions; and elevates the consumption of the reactant by side-reactions with water (Fig. 3b). Such selective esterification of cellulose was so far only possible in the dissolved form, i.e. under homogeneous conditions, using bulky ester groups[55] or by complex multi-step syntheses, especially in the case of small ester groups, such as acetyl[56]. Therefore, apart from the confinement of the reaction system by surface water, which is a more theoretical facet of our work, here we describe a direct method to esterify the C6–OH of the cellulose fibre surface with high selectivity under solid-state conditions. Going beyond esterifications and derivatizations, only an oxidative heterogeneous method, the TEMPO-oxidation, has been able to selectively modify the surface C6–OH[57] with regioselectivities similar or only slightly superior to our approach.

Chemical-mechanical reactions under ball milling[50] as well as conventional acid-catalysed heterogeneous surface acetylations[58] have been reported to influence cellulose's bulk properties, reducing its crystallinity and molecular weight. These effects are highly undesirable as they will reduce the materials performance[59–61], especially with regard to mechanical properties. Importantly, the reported confined surface acetylation did not affect the bulk crystallinity but increased the molecular weight of cellulose (Supplementary Fig. 1). This is, evidently, in contrast

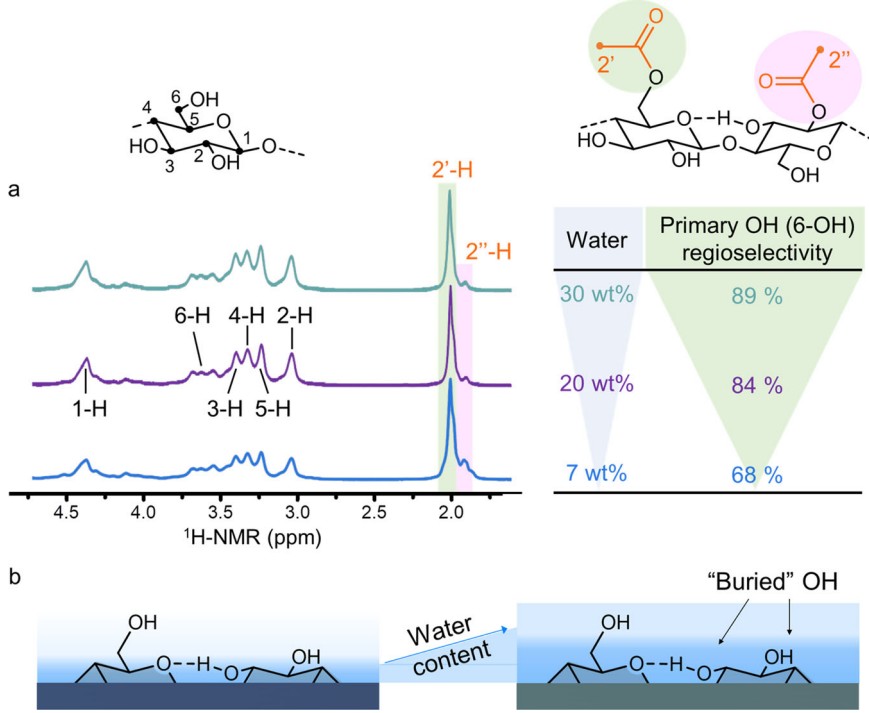

**Fig. 3 Water-mediated control of regioselectivity.** Diffusion-edited [1]H-NMR spectra (**a**) identified that increasing water allowed for more control over the acetylation regioselectivity. The primary hydroxyl group (6-OH) was modified with up to 89% selectivity in the presence of 30 wt% water, unparalleled for a simple esterification. At elevated water content and higher hydration, the secondary hydroxyl groups are 'buried' rendering them chemically less accessible (**b**).

to all methods employing homogeneous derivatization, especially the well-studied acid-catalysed heterogeneous industrial methods for production of cellulose acetate[36] but also the well-known TEMPO oxidation[62]. We further compared the surface structure of acetylated fibres and native fibres with electron and light microscopy (Fig. 4a–c and Supplementary Fig. 4a, b). Independent from the introduced degree of acetylation, the cellulosic fibre structure was hardly affected and is hence well preserved (only minor fibrillation occurred at the fibre surface). As the reaction was performed under heterogeneous conditions, we studied the fibre wettability and surface chemistry by Raman mapping to prove a homogeneous surface coverage. The wettability of the fibres was influenced dramatically, and the water contact angle increased from 12° up to ~134° for the acetylated fibres (Supplementary Fig. 4d), indicating a pronounced fibre hydrophobicity. Chemical imaging of the fibre surface by Raman further confirmed a homogeneous coverage of the surface by acetyl groups, even though we conducted our acetylation in the solid state (Fig. 4e, f and Supplementary Fig. 4c). This is another evidence supporting the involvement of uniform hydration layers as the reaction medium.

**Reaction efficiency and role of water.** Acetylation at EMC yielded a DS of 0.49 ± 0.04 corresponding to the theoretical number of accessible C6- and C2-hydroxyls of ~0.5 mmol/mmol (Supplementary Table 1). The acetylation at C3 position seems to be negligible under our conditions; this is consistent with the literature, since the C3–OH is less accessible and hardly reactive due to its intra-molecular hydrogen bond with the cyclic oxygen atom of the neighbouring cellulose monomer unit[63,64] (see chemical structure in Fig. 2a). We further studied acetylation using lower equivalents of *N*-acetylimidazole (0.3 eq.), demonstrating a reaction efficiency of up to 80% (at EMC) (Supplementary

Table 2). This efficiency is remarkable for a bulk solvent-free and heterogeneous reaction (but still containing significant surface water). It is significantly higher than the solvent-based acetylation of cellulose with *N*-acetylimidazole[44] and comparable to the most efficient acetylations of cellulose under homogeneous conditions[65,66]. While increasing the amount of *N*-acetylimidazole decreases the reaction efficiency and the reaction rate (Supplementary Table 2 and Supplementary Fig. 5), it seems that the increased reaction time and amount of *N*-acetylimidazole elevates the frequency of side reactions. This includes hydrolysis of *N*-acetylimidazole, as a major side reaction, which is also imidazole-catalysed[45]; and depletes our reaction system of its 'foundation' through water consumption.

We further studied the influence of water in a reaction with 0.3 equivalents of *N*-acetylimidazole and obtained similar results as in the case of 1.5 equivalents: The presence of a water hydration layer (EMC) led to significantly higher reaction kinetics than the dry fibre (dry, ~0 wt% moisture content) (Supplementary Fig. 6). We noticed that a sequential acetylation using 0.3 equivalents *N*-acetylimidazole at EMC was not possible: We could not acetylate further a fibre with a DS of 0.2, revealing that a native cellulose surface (with its intrinsic hydration layer) was mandatory to enable the reaction.

Additionally, we compared the results to an experiment with a water-free but DMSO-wetted fibre (using the same volume of DMSO as water in the fibre at EMC); this experiment was chosen to reveal if water is directly involved in the reaction mechanism or has only physical effects, such as the solvation of the reactant or a swelling effect to increase the fibres' accessibility. As shown in Supplementary Fig. 6, the presence of DMSO increased the reaction rate in comparison to the dry fibre, indicating that swelling was important. However, although *N*-acetylimidazole is freely soluble in DMSO[67] and DMSO is a strong cellulose swelling agent[68], the reaction rate in the case of the DMSO-fibre

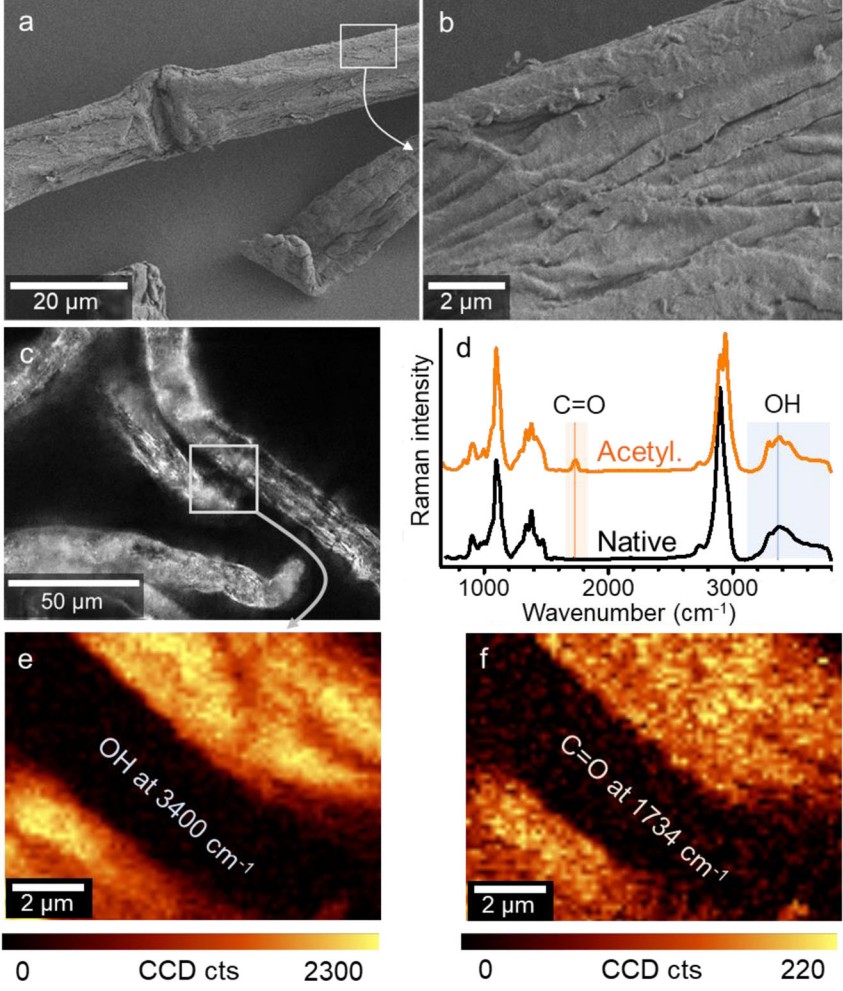

**Fig. 4 Imaging of the surface of acetylated fibres.** The confined water-promoted acetylation preserved the morphology of the native cellulosic fibres, as demonstrated via scanning electron (**a**, **b**) and light microscopy (**c**). The carbonyl (C=O) band of the acetyl group is clearly shown in the Raman spectrum of the acetylated fibre (**d**). Raman mapping images of the hydroxyl (OH) and C=O band (**e** and **f**) prove a homogeneous distribution of the acetyl groups on the cellulosic fibre. Raman signal intensities in counts (cts), measured with a charge-coupled device (CCD) detector, are represented by the colour scales.

was still lower than that of the moist fibre at EMC. This indicates that the surface-confined water is not only acting as solvent medium, increasing the swelling, and thus diffusion of the reactant in the fibre, but it is also directly implicated in the chemical reaction mechanism. It is reported that water is involved in proton transfer reactions from imidazole groups[45,69]. We assumed that water, as protic solvent, facilitates the proton transfer in the transition state from the cellulose acyl imidazolium intermediate (as schematically shown in Fig. 5a) to the imidazole base catalyst.

A possible transition state involving a proton transfer of water in a concerted reaction mechanism is shown in Fig. 5a and was further validated through a gas-phase computational study of the acetylation of methanol (as the simplest alcohol model compound) (Fig. 5c). Without water: the Gibb's free energy of activation ($\Delta G^{TS}$) was approximated to 28 kcal/mol. Addition of water relieved the strain in the transition state geometry and, therefore, reduced tremendously $\Delta G^{TS}$ to 16 kcal/mol (Fig. 5c, d). Clearly having a small amount of water in the reaction is advantageous for esterification kinetics (due to reduced strain during proton transfer). The mechanism, with water, was also modelled on a cellulose Iβ surface fragment for comparison with the methanol model (Fig. 5e): The reaction proceeded in an

analogous mechanism with a very similar $\Delta G^{TS}$ of ~15 kcal/mol (see Supplementary Section 1 for more details).

**Discussion**

Cellulosic fibres, as well as other nanoporous renewable biopolymers, are hygroscopic and intrinsically confine water in its fibrillar structure. Our data show, that such confined water can be exploited as nanomedium to promote singular chemical reactivity at the biopolymer's surface.

In comparison to a dry acetylation, surface-confined water elevates the reaction efficiency and increases the reaction rate of the cellulose surface acetylation with *N*-acetylimidazole. In addition, the chemical accessibility of the cellulose hydroxyl groups can be controlled by the amount of confined water on the nanofibre surface, enabling the largely regioselective surface acetylation of cellulose. Modelling of the reaction transition states reveals that the presence of water reduces the activation barrier of the acetylation, by avoiding constrained geometries and enabling a concerted proton transfer mechanism. This, evidently, is in stark contrast to the unfavourable interference of bulk water with conventional esterifications.

More broadly, we demonstrate that confining reactions inside the hydration layer of nanoporous materials is a feasible concept

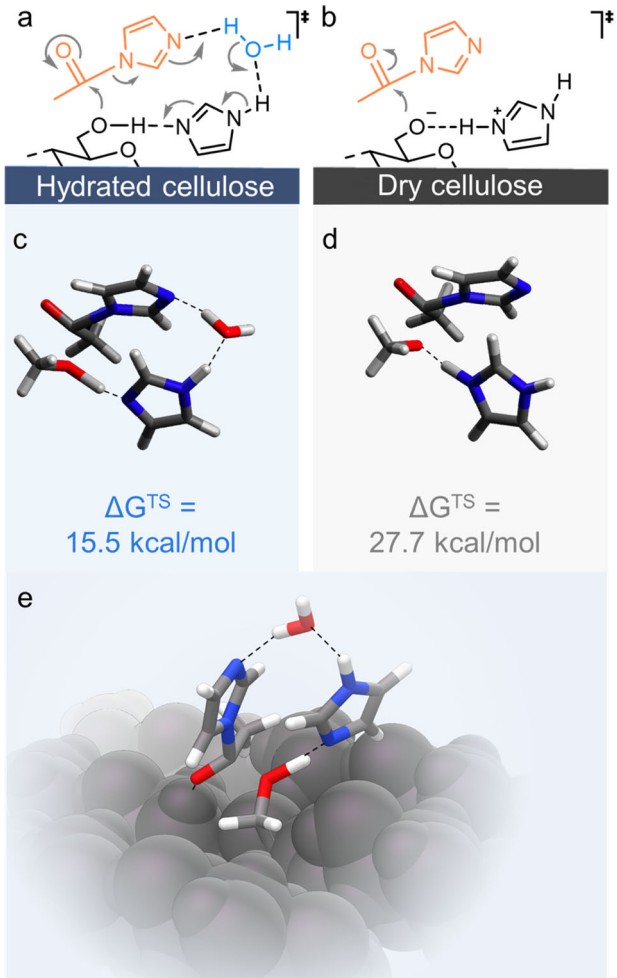

**Fig. 5 Mechanism of the confined water-promoted surface reaction.**
Schematic representation of the possible involvement of the surface-confined water in the imidazole-catalysed acetylation with *N*-acetylimidazole (**a**, **b**). As calculated using methanol as model primary alcohol, presence of water significantly reduced the Gibb's free energy of activation ($\Delta G^{TS}$) (**c**, **d**). An analogue transition state was modelled on a cellulose fragment (**e**). The presence of water facilitates proton transfer promoting the acetylation in a concerted mechanism. Significant strain of the geometry is required to get proton transfer, in the absence of water.

to promote reactions at the polymer surface. We believe that this work can be adapted to a broad range of polymers and reactions and will lead to significant advances in the general field of materials chemistry, analytics and catalysis.

## Methods
The cellulosic fibres, TCF-bleached beech sulphite dissolving-grade pulp, were supplied by Lenzing AG (Austria) with a moisture content of ~7 wt%. Properties and specification of the pulp are listed in Supplementary Table 1. The used chemicals were, if not otherwise noted, purchased from Sigma-Aldrich (Sigma-Aldrich Chemie GmbH, Munich, Germany) with a purity of at least 99%. *N*-Acetylimidazole was supplied in a purity of ≥98% from TCI (TCI Deutschland GmbH, Germany).

**Pre-treatment**. Imidazole and cellulosic fibres were pre-treated before the solid-state acetylation. The imidazole flakes were pulverised for 2 min in a coffee grinder (KSM 2 Grinder 4041, Braun GmbH, Germany) and stored under exclusion of moisture. The cellulose fibres (2 g) were pre-treated in a vibratory ball mill (Retsch CryoMill, Retsch GmbH, Germany) for 15 min at 25 Hz in a 50 mL vessel using five grinding balls (stainless steel) with a diameter of 0.5 cm and one with a diameter of 1 cm to obtain fluffy fibres (this pre-treatment only increased the initial reaction kinetics and is not mandatory, see Supplementary Fig. 7).

**Sample conditioning**. The fluffy fibres were equilibrated at a relative humidity (RH) of 50% and 20 °C, the EMC at 50% RH (EMC) was determined gravimetrically to be ~7 wt%. This sample was used for the experiments conducted with fibres at EMC. The dry sample was obtained by vacuum-drying at 40 °C and 10 mbar for 48 h. The dry sample was used to prepare the fibres at 20 and 30 wt% by mixing and equilibrating the fibres in a closed vessel at 20 °C with the respective amount of water for at least 24 h (Samples 20 and 30 wt%). The DMSO sample was prepared by equilibrating the dry sample with dry DMSO (using the same volume as the moisture in the EMC sample, i.e. 7% v/w) in a closed vessel for at least 24 h at 20 °C.

**Solid-state acetylation with *N*-acetylimidazole**. Cellulose fibres at EMC (1.0 g dry mass, 6.2 mmol, 1 Eq) were mixed with the respective amount of *N*-acetylimidazole, i.e. 0.3 Eq (0.20 g, 1.85 mmol) or 1.5 Eq (1.02 g, 9.25 mmol), in a 0.25 mL grinding jar. 10 mol% imidazole (based on the amount of *N*-acetylimidazole) and stainless-steel grinding balls (17.7 g, 0.5 cm diameter) were added and milled for 0.5 h at 25 Hz in a vibratory ball mill (Retsch CryoMill, Retsch GmbH, Germany). After milling, the samples were equilibrated at 20 °C in a closed vessel. The reaction time was defined as the sum of the milling time and equilibration times. The acetylation was quenched by addition of a saturated aqueous solution of NaHCO₃. The acetylated fibres were finally washed carefully with DI water and dried at 105 °C for further analysis. The acetylation was performed at various conditions as listed in Supplementary Table 2 to study the influence of confined water and the amount of *N*-acetylimidazole on the fibre acetylation (all reactions were performed in closed vessels at 20 °C). The DS (or degree of acetylation) was determined using infra-red spectroscopy in attenuated total reflection mode (PerkinElmer Frontier IR single-range spectrometer (PerkinElmer Inc., USA)) using the calibration curve in Supplementary Fig. 8. Further details can be found in the Supplementary Methods.

## Data availability
Data underlying reported averages in tables and figures, the calibration curve, and the calculation of the regioselectivity is available in figshare repository at https://doi.org/10.6084/m9.figshare.14096071.v1. Other data is available from the corresponding authors upon reasonable request.

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

## Acknowledgements

We wish to acknowledge CSC—IT Centre for Science, Finland, and Finnish Grid and Cloud Infrastructure (persistent identifier urn:nbn:fi:research-infras-2016072533) for computational resources. We thank Dr. Markus Bacher, Dr. Sonja Schiehser and Dr. Caio G. Otoni for their support with NMR, GPC and DVS measurements, respectively. Dr. Blaise L. Tardy is acknowledged for helpful discussions and SEM measurement. This research was funded in whole, or in part, by the Austrian Science Fund (FWF) (J4356). For the purpose of open access, the author has applied a CC BY public copyright license to any Author Accepted Manuscript version arising from this submission. The authors thank the financial support from the Austrian Biorefinery Centre Tulln (ABCT), the Academy of Finland (Project # 311255, 'WTF-Click-Nano') and the H2020-ERC-2017-Advanced Grant 'BioELCell' (788489).

## Author contributions

M.B. and T.R. conceived the idea and supervised the project. P.J. and M.B. contributed to general materials preparation and characterisation. N.G. contributed to structural characterisation with Raman and light microscopy. A.W.T.K. performed NMR analysis and computational calculations. A.W.T.K., A.P., O.J.R., T.R. and M.B. provided important conceptual insights. All authors contributed to the interpretation of the results. M.B., A.W.T.K. and P.J. wrote the paper in consultation with and input of all authors.

## Competing interests

The authors declare no competing interests.
