## [Peer Review File · Nature Communications]

REVIEWER COMMENTS

Reviewer #1 (Remarks to the Author):

This paper summarizes the experimental results regarding the reactivity with water contained on the surface of cellulose having a nanoporous structure, and elucidates the reaction efficiency improvement mechanism by the contained water using simulation. The results obtained by the experiment are clear, and there is no major discrepancy in the consideration from the results of the simulation analysis, and it is judged that the content is worth publishing in this journal. However, some questions arose. I will leave my comment below.

In this paper, discussions are being developed regarding cellulose as a nanoporous material. Fig. 1 shows a schematic diagram of cellulose. Where is the part of this diagram that can be regarded as nanoporous? Also, do you think that the nanoporous here has connectivity? Adsorption of water is a very important content in this paper, and how much water permeates is considered to be important information in this paper. Please make corrections by adding this information.

In this paper, acetylation in the solid state is performed using a ball mill. Is this method using a mechanochemical reaction? In that case, please show the superiority of this paper in terms of reaction speed compared to the previous literature. Also, could you add detailed test conditions? In particular, the test environment during a ball mill, specifically humidity and temperature, is judged to be important information for showing reproducibility.

Some data in Fig. 13 is missing. Please correct the figure displayed correctly.

Reviewer #2 (Remarks to the Author):

This manuscript describes mechanochemically assisted heterogeneous acetylation of cellulose. The authors have made many elegant experiments to reveal how the “non-freezing bound water” affects heterogeneous acetylation reaction of surface-associated hydroxyl groups. The key merit of this study is the new mechanism for the confined water-promoted surface-restricted acetylation of cellulose with N-acetylimidazole in the presence of imidazole. I believe the results are of broad importance and may trigger related studies on other natural polymers. In this sense, the present study lays important foundation in the field. However, my main trouble with the present form of the manuscript relates to the somewhat unclear description of the drying and wetting procedures as well as incomplete discussion of the swelling and reagent accessibility aspects that I believe are central to building a coherent picture of the phenomenon. I therefore support publication of this study in Nature Communications provided that the authors address my critical comments as detailed below.

1. The title appears overselling since no other chemical reactions than that of cellulose acetylation were demonstrated. I suggest re-writing the title to reflect the actual work conducted. Keywords for the title: cellulose, acetylation, surface-bound water, high reactivity... This will be helpful for indexing the paper once published.
2. Previous studies (e.g. DOI: 10.1016/j.carbpol.2010.10.016) have shown that acetylation proceeds from surfaces to the core fraction of cellulosic fibres. Taking into account the porosity of the pulp

fibres, to what extent the acetylation was restricted on the surfaces in the present study?

3. Swelling of cellulose is well-known to affect its reactivity due to changes in accessibility and interfibrillar interactions. These differences should be quantified and their impact on the rate of the acetylation reaction shown in Figure 2c elucidated. Further, in the experimental section it is mentioned that EMC contained 7% moisture, but the vacuum oven-dried batch of cellulose was used to prepare samples at various moisture contents. It is unclear to the reviewer if EMC was also prepared from vacuum-dried cellulose or it was the never-dried sample equilibrated at 50% RH. This question even extends to the DMSO-equilibrated cellulose. The origin of the samples should be clarified and the extent of swelling and accessibility of the samples determined using quantitative methods such as dye adsorption, solute exclusion, porosimetry, or other.

4. How does the water content of cellulose affect solubilisation of N-acetylimidazole and in turn the kinetics of the acetylation reaction? Compare this to the reactivity in the dry state.

5. Literature review should be strengthened. Mechanochemical acetylation of cellulose is not itself a novel approach. See for instance DOI: 10.1016/j.compscitech.2008.05.005. The state of the art of cellulose-water interactions especially based on water uptake studies should be cited and discussed.

6. The illustrations need improvements. Figure 1: the counter clockwise representation of the study approach could be improved in clarity and flow. Figure 2: include hydrogen bonding between surface-bound water and cellulose hydroxyls. Figure 3B: use different colours to improve contrast between “buried OH” and the background (now both in blue).

7. A few excerpts requiring rephrasing or clarification:

“Aligning our reasoning with previous studies of the acetylation of tyrosine in enzymes, stating that certain “buried” tyrosine residues are not chemically accessible due to hydrogen bonding.⁴¹”  This sentence is not well-connected to the surrounding discussion.

“Therefore, apart from the confinement of the reaction system by surface water, which is a more theoretical facet of the work, this report describes the first direct method to esterify the C6-OH of the cellulose fibre surface with high selectivity under sustainable solid-state conditions.”  Please define “sustainable”, preferably using sustainability metrics such as atom-efficiency etc.

8. Minor corrections:

- Page 10: Figure 4A-E should refer to Figure 5A-E respectively.

- Figure S8: please indicate in the figure caption the wavenumbers used to calculate the absorbance intensity ratio used for the calibration curve.

Mika H. Sipponen

Dear esteemed reviewers,

We greatly appreciate the comments of the referees of our manuscript entitled “*Unique Reactivity in Surface-Confined Water of Nanoporous Polymers*”, which is under consideration for publication in **Nature Communications** (NCOMMS-20-40389).

All the comments are addressed in the revised version of the manuscript. In the revised files, the changes are highlighted in yellow colour (see MARKED version of the manuscript). Additionally, UNMARKED versions of the revised files are provided. We appreciate very much the time and suggestions provided by the expert reviewers. Their feedback helped to enhance the quality and impact of our contribution.

Please find the answer to the reviewer comments below:

REVIEWER #1 This paper summarizes the experimental results regarding the reactivity with water contained on the surface of cellulose having a nanoporous structure and elucidates the reaction efficiency improvement mechanism by the contained water using simulation. The results obtained by the experiment are clear, and there is no major discrepancy in the consideration from the results of the simulation analysis, and it is judged that the content is worth publishing in this journal. However, some questions arose. I will leave my comment below.

Authors: Thank you, we really appreciate the positive feedback. We have edited the manuscript according to each and all the comments.

Comment 1: In this paper, discussions are being developed regarding cellulose as a nanoporous material. Fig. 1 shows a schematic diagram of cellulose. Where is the part of this diagram that can be regarded as nanoporous? Also, do you think that the nanoporous here has connectivity? Adsorption of water is a very important content in this paper, and how much water permeates is considered to be important information in this paper. Please make corrections by adding this information.

Authors 1: This is a very reasonable comment, and we agree that the points raised require clarification. The nanoporous structure of cellulosic fibres originates from its interfibrillar structure (densest packing is 0.9069). We modified Fig. 1 to better visualize the interfibrillar distances (pores) and added further clarification in the caption of Fig.1. The periodic structural openings enable interactions between water, or other small molecules, with the inner structures of nanofiber bundles. The swelling of cellulosic materials arises from the diffusion of water molecules in between the primary building blocks (the smallest nanofiber unit, *i.e.* elementary fibril). We further clarify these points in the introduction section on Page 3.

As discussed in the paper and given the fact that the theoretically accessible hydroxyls were modified, our results show that the surfaces of the elementary fibrils are accessible to water. This further confirms the interconnectivity of the pores - here it is important to consider that such interconnectivity might be closely related to environmental conditions (e.g., humidity or solvent).

The swelling properties of the fibre were investigated following the evaluation of the fibre saturation point, water retention value, specific pore surface, average pore diameter, pore volume (Table S1), and water vapor sorption (Figure S10).

Comment 2: In this paper, acetylation in the solid state is performed using a ball mill. Is this method using a mechanochemical reaction? In that case, please show the superiority of this paper in terms of reaction speed compared to the previous literature. Also, could you add detailed test conditions? In particular, the test environment during a ball mill, specifically humidity and temperature, is judged to be important information for showing reproducibility.

Authors 2: The method used in our work is in clear contrast to conventional (mechano)chemical esterification: the acetylation occurred spontaneously and did not require continuous energy input, as otherwise is supplied in typical milling. Our short milling was used just to achieve efficient mixing. We clarify this in the manuscript by pointing to the fact that heterogeneous reaction of cellulose and *N*-acetylimidazole commenced spontaneously upon solid-state mixing. Also, in contrast to conventional (mechano)chemical esterifications, no continuous energy input in the form of milling or heating was required as the acetylation occurred spontaneously. Finally, the kinetics of our solid-state reactions were very fast at room temperature; this contrasts with conventional mechanochemical esterification of cellulose fibres, which requires 15 h of continuous milling to achieve similar rates as those shown in our approach (DOI 10.1002/cssc.201200492)!

All reactions were performed in closed vessels at 20 °C and the used chemicals were fresh and anhydrous; hence, the water in the system originated only from the fibre's moisture (the air in the 25 mL-vessel can be neglected). We added information related to temperature and reactions conditions in the revised manuscript (Experimental Section).

Comment 3: Some data in Fig. 13 is missing. Please correct the figure displayed correctly.

Authors 3: According to this observation, we replaced Figure S13 and S14 (in the previous version they showed different spectral width compared to the other spectra, and the upper region was cut in Figure S13).

REVIEWER #2:

This manuscript describes mechanochemically assisted heterogeneous acetylation of cellulose. The authors have made many elegant experiments to reveal how the “non-freezing bound water” affects heterogeneous acetylation reaction of surface-associated hydroxyl groups. The key merit of this study is the new mechanism for the confined water-promoted surface-restricted acetylation of cellulose with N-acetylimidazole in the presence of imidazole. I believe the results are of broad importance and may trigger related studies on other natural polymers. In this sense, the present study lays important foundation in the field. However, my main trouble with the present form of the manuscript relates to the somewhat unclear description of the drying and wetting procedures as well as incomplete discussion of the swelling and reagent accessibility aspects that I believe are central to building a coherent picture of the phenomenon. I therefore support publication of this study in Nature Communications provided that the authors address my critical comments as detailed below.

Authors: We appreciate the positive and insightful feedback provided by the reviewer. Following the suggestion, we have added a more detailed description of the drying and wetting protocols and expand the discussion about the swelling and chemical accessibility. We also wish to clarify, as was pointed to Reviewer #1 (comment 2) and also, in light of question # 5 below, that our esterification did not involve, at least to any appreciable extent, mechanochemical reactions.

Comment 1: The title appears overselling since no other chemical reactions than that of cellulose acetylation were demonstrated. I suggest re-writing the title to reflect the actual work conducted. Keywords for the title: cellulose, acetylation, surface-bound water, high reactivity... This will be helpful for indexing the paper once published.

Authors 1: We concur with the opinion of the reviewer and have changed the title to: “Singular Reactivity of Nanoporous Materials by Confined Water: Enhanced and Selective Surface-Acetylation of Cellulosic Fibres”.

Comment 2: Previous studies (e.g. DOI: 10.1016/j.carbpol.2010.10.016) have shown that acetylation proceeds from surfaces to the core fraction of cellulosic fibres. Taking into account the porosity of the pulp fibres, to what extent the acetylation was restricted on the surfaces in the present study?

Authors 2: This is a very valuable question. The previous work cited by the reviewer considered acetylation with acetic anhydride and iodine, similar to H₂SO₄-catalyzed acetylation. These reactions are challenging to confine only to the surface of the individual nanofibers (or microfibrils). Usually, the reactions proceed into non-accessible regions (more ordered or crystalline core of the nanofibers), eventually reducing crystallinity and molar mass and, as result, undermining the mechanical properties (as is also shown in the cited paper, 10.1016/j.carbpol.2010.10.016).

In contrast to conventional approaches, our method is heterogeneous, confined to the accessible cellulose regions (nanofiber surface) and non-degrading, which we prove through solid-state NMR and GPC measurements. Noteworthy, independently of reaction time and conditions, the crystallinity and molar mass remained unaffected. We would like to draw the attention to the fact that nanofiber core (chemically non-

accessible, crystalline regions) is different than the micron-sized fibre core. The latter is a definition related to local nanofiber assemblies, of larger characteristic dimensions.

Since we used fibres as our starting material, the modification occurred *via* diffusion of the reactants through the hydration layers on the fibres. Principally, we propose that nanofiber surfaces situated at the microfiber surface were first modified, and then the reaction proceeded toward the microfiber core (still modifying only the nanofiber surfaces) and limited by diffusion. This is discussed in the context of Figure S2 and also shown by Raman and contact angle analyses of the samples with a DS of 0.3 and 0.5 (Figure S4). Despite the fact that these samples had different DS, they both showed similar C=O Raman intensities and water contact angles, indicating full microfiber surface acetylation.

Comment 3: Swelling of cellulose is well-known to affect its reactivity due to changes in accessibility and interfibrillar interactions. These differences should be quantified and their impact on the rate of the acetylation reaction shown in Figure 2c elucidated. Further, in the experimental section it is mentioned that EMC contained 7% moisture, but the vacuum oven-dried batch of cellulose was used to prepare samples at various moisture contents. It is unclear to the reviewer if EMC was also prepared from vacuum-dried cellulose or it was the never-dried sample equilibrated at 50% RH. This question even extends to the DMSO-equilibrated cellulose. The origin of the samples should be clarified and the extent of swelling and accessibility of the samples determined using quantitative methods such as dye adsorption, solute exclusion, porosimetry, or other.

Authors 3: We start with the latter point raised by the reviewer to indicate that the origin of the samples is presented in the Experimental Section (sub-section about pre-treatment of the previous version of the manuscript). The revised manuscript includes a new section “Sample conditioning” that was added to address the point raised by the reviewer, that explains the pre-conditioning and sample preparation in detail. All fibre specimens were prepared from once-dried cellulose pulp, which was equilibrated at a relative humidity of 50% and 20 °C. EMC samples were directly used from this sample. All other samples were vacuum-dried and then equilibrated to the respective conditions as stated in more detail in the Experimental Sub-section.

We used a standard dissolving-grade pulp fibre to ensure high cellulose content and to reduce any influence of other wood components, such as hemicelluloses and lignin. The swelling characteristics and water accessibility of these fibres and their properties are comprehensively characterized, as summarized in Table S1 (including information on composition, molar mass, crystallinity, specific and pore surface area, fibre saturation point, water retention value and pore volume).

Comment 4: How does the water content of cellulose affect solubilisation of N-acetylimidazole and imidazole and in turn the kinetics of the acetylation reaction? Compare this to the reactivity in the dry state.

Authors 4: The solubility of acetylimidazole (AI) is low, approx. 50 mg/mL, implying that at a moisture content of 7 wt%, only 3.5 mg could be dissolved (from 204 mg in the case of 0.3 Eq of AI or from 1020 mg in the case of 1.5 Eq of AI). This indicates that the reaction occurred under conditions of nano-solvation, inside the hydration layer, and that solubility played a minor role. A higher solubility did not improve the

reaction rate, as shown in the with DMSO results (Figure S6, noting that the solubility of AI in DMSO is ca. 330 mg/mL). At the starting condition, approx. 23 mg of AI can be dissolved in DMSO (10 times the amount compared to that in the case of water); however, the reaction was slower than in water. A higher amount of water, which can dissolve more AI, reduces reaction efficiency and does not increase reaction rate (Figure S3B).

Comment 5: Literature review should be strengthened. Mechanochemical acetylation of cellulose is not itself a novel approach. See for instance DOI: 10.1016/j.compscitech.2008.05.005. The state of the art of cellulose-water interactions especially based on water uptake studies should be cited and discussed.

Authors 5: We agree with the reviewer but would like to clarify that our procedure cannot be classified as traditional mechanochemical acetylation (see also above). We added additional literature and further clarified this in the manuscript:

Page 4: “Efficient mixing was realized by using short vibratory ball milling. In contrast to conventional (mechano)chemical esterifications,^{47–49} acetylation occurred promptly, with no need for continuous energy supply in the form of milling (and heating). Earlier reports on (mechano)chemical esterifications of cellulose and other polymers have relied on the presence of liquid/solution media (chemicals), such as acetic anhydride. To the best of our knowledge, our presented approach is the only true solid-state polymer acetylation method available thus far.”

We also added a section dedicated to cellulose-water interactions in the Introduction. Page 3: “The cellulose nanofibres are composed of inter-connected, axially twisted crystallites²⁵ with a chemically accessible and hydrophilic surface.²⁶ The nanoporosity of cellulose originates from this interfibrillar assembly and structural defects that arise from the spatial mismatch and dislocations of individual nanofibers and assembled bundles, enabling solvent interpenetration and fibre swelling.²⁷ The abundance of surface hydroxyl groups in this hierarchical structure endows hygroscopicity and sorption-induced swelling of cellulose fibres.^{28,29} Main contributors in the swelling are the surfaces of the individual nanofibers,^{30,31} that are covered with hydration layers, under ambient conditions^{26,28}. Due to the strong cellulose-water interactions, this surface-bound water can be distinguished from bulk water by its properties and it is referred to as freezing or non-freezing (bound) water^{32–35}. Non-freezing water is directly situated at the nanofibre surface, whereas freezing water accumulates in interfibrillar pores.³²”

Comment 6: The illustrations need improvements. Figure 1: the counter clockwise representation of the study approach could be improved in clarity and flow. Figure 2: include hydrogen bonding between surface-bound water and cellulose hydroxyls. Figure 3B: use different colours to improve contrast between “buried OH” and the background (now both in blue).

Authors 6: Thank you for this remark, Fig. 1, and Fig. 3B were optimized and redrawn to improve their quality and clarity.

Comment 7: A few excerpts requiring rephrasing or clarification: “Aligning our reasoning with previous studies of the acetylation of tyrosine in enzymes, stating that

certain “buried” tyrosine residues are not chemically accessible due to hydrogen bonding.⁴¹  This sentence is not well-connected to the surrounding discussion.

“Therefore, apart from the confinement of the reaction system by surface water, which is a more theoretical facet of the work, this report describes the first direct method to esterify the C6-OH of the cellulose fibre surface with high selectivity under sustainable solid-state conditions.”  Please define “sustainable”, preferably using sustainability metrics such as atom-efficiency etc.

Authors 7: We appreciate the opportunity to improve the text and to make it easier to understand– we have modified the respective sections and also further improved the whole manuscripts (see highlighted changes in the manuscript).

Comment 8: Minor corrections:

- Page 10: Figure 4A-E should refer to Figure 5A-E respectively.
- Figure S8: please indicate in the figure caption the wavenumbers used to calculate the absorbance intensity ratio used for the calibration curve.

Authors 8: The Figure numbers were updated, and we added the corresponding information in the caption of Figure S8.

Additional changes

- Further proof-reading of the manuscript (see highlighted changes).
- Figure 5 was rearranged to improve clarity.
- Figures S3, S6 were updated: Sample name “0%” was changed to “dry” for better consistency
- Minor corrections in Table S2, averages and standard deviations were recalculated.
- Analysis and calculation of the acetylation regioselectivity was added as subsection into Supplementary Methods.
- Source data underlying figures, calculation of regioselectivity and the calibration curve are available (cf. Data Availability section in the main text).

REVIEWERS' COMMENTS

Reviewer #1 (Remarks to the Author):

I have determined that the content of the revised treatise has been properly modified for the question. The comments from other reviewers have also been answered accurately, and I have decided that this paper can be published in this journal.

Reviewer #2 (Remarks to the Author):

The authors have adequately addressed all of my remarks and improved the manuscript in clarity. I support publication of the revised version Nature Communications.

Mika H. Sipponen